# Optimized Sugar-Free Citrus Lemon Juice Fermentation Efficiency and the Lipid-Lowering Effects of the Fermented Juice

**DOI:** 10.3390/nu15245089

**Published:** 2023-12-13

**Authors:** Chang-Lu Hsu, Wen Pei, Tzu-Chun Chen, Ming-Chieh Hsu, Pei-Chun Chen, Heng-Miao Kuo, Jeng-Fung Hung, Yi-Jinn Lillian Chen

**Affiliations:** 1College of Management, Chung Hua University, Hsinchu 30012, Taiwan; d11003008@chu.edu.tw (C.-L.H.); wpei@chu.edu.tw (W.P.); 2Graduate Institute of Science Education & Environmental Education, National Kaohsiung Normal University, Kaohsiung 82444, Taiwan; ulusai@hotmail.com (T.-C.C.); vivianjay79@gmail.com (P.-C.C.); duncan750409@gmail.com (M.-C.H.); t1873@nknu.edu.tw (J.-F.H.); 3Institute of Phytochemicals Jianmao Biotech Co., Ltd., Kaohsiung 80672, Taiwan; susan.kuo@twxlife.com; 4Department of Physics, National Kaohsiung Normal University, Kaohsiung 82444, Taiwan

**Keywords:** efficiency, hyperlipidemia, fermented lemon juice, fermentation evaluation, lipid-lowering

## Abstract

Aging and obesity make humans more prone to cardiovascular and metabolic syndrome diseases, leading to several serious health conditions, including hyperlipidemia, high blood pressure, and sleep disturbance. This study aimed to explore the hypolipidemic effect of fermented citrus lemon juice using a hyperlipidemic hamster model. The sugar-free lemon juice’s fermentation was optimized, and the characteristics of fresh and fermented lemon juice (FLJ) were evaluated and compared, which contained polyphenols and superoxide dismutase-like activity. Results showed that the absorption and utilization efficiency of FLJ was higher compared with the unfermented lemon juice. This study’s prefermentation efficiency evaluation found that 21–30 days of bacterial DMS32004 and DMS32005 fermentation of fresh lemon juice provided the best fermentation benefits, and 21-day FLJ was applied as a remedy after the efficiency compassion. After six weeks of feeding, the total cholesterol (TC) and triglyceride (TG) values in the blood and liver of the FLJ treatment groups were decreased compared with the high-fat diet (HFD) group. In addition, the blood low-density lipoprotein cholesterol (LDL-C) levels were significantly reduced in the FLJ treatment groups compared with the HFD group. In contrast, the blood high-density lipoprotein (HDL-C) to LDL-C ratio increased considerably in the FLJ treatment groups, and the total to HDL ratio was significantly lower than in the HFD group. Compared with the HFD group, the TC content in the FLJ treatment groups’ feces increased significantly. This study demonstrated that the sugar-free fermentation method and fermentation cycle management provided FLJ with the potential to regulate blood lipids. Further research and verification will be carried out to isolate specific substances from the FLJ and identify their mechanisms of action.

## 1. Introduction

Lemon is a popular citrus fruit that contains bioflavonoids and other bioactive compounds, such as phenolic compounds, organic acids, essential oils, vitamins, carotenoids, pectin, and minerals. Lemons are recognized to prevent diseases and have anticancer, antimicrobial, and lipid-lowering effects. Furthermore, lemons have been demonstrated to have a protective effect against cardiovascular diseases [1].

According to the World Health Organization (2021) [2], about 40% of the world’s population is obese, and the epidemic of childhood and adult obesity is growing. Obesity can cause serious health problems and increase the risk of heart and circulatory disease, including dyslipidemia, hypertension, and sleep disorders. Hyperlipidemia means that blood has too many lipids, which is characterized by elevated plasma levels of total cholesterol (TC), triglycerides (TGs), and low-density lipoprotein cholesterol (LDL-C), which is ‘bad cholesterol’ that clogs arteries. Cholesterol, particularly low-density lipoprotein cholesterol, triglycerides, and other fats can build up in the arteries, narrowing the blood vessels and making it harder for blood to pass. In contrast, high-density lipoprotein cholesterol (HDL-C) is protective against heart and blood vessel diseases because it absorbs cholesterol in the blood and takes it to the liver, which eliminates it from the body [3,4,5]. 

Studies suggest that the intake of citrus fruits and their juices can prevent cardiovascular disease, which may be related to citrus bioflavonoids [6,7,8]. Citrus bioflavonoids are a class of antioxidant compounds, such as naringenin, hesperidin, nobiletin, and tangerine [9,10]. For example, hesperidin has been demonstrated to improve blood lipid regulation in animals with casein-induced hyperlipidemia [11,12]. In addition, hesperidin metabolites improve blood lipid levels, which are speculated to be related to the decreased activity of cholesterol synthase and esterase [13,14,15]. 

Previous animal studies have demonstrated that fermentation improves the bioavailability of substances, such as flavanols, in orange juice [12]. In studies on the effect of fermentation on the absorption rate of polyphenols in humans, the results showed that the polyphenols in fermented orange juice were absorbed more rapidly after ingestion [16,17]. 

Unlike rats and mice, which are often used in experiments, hamsters have lipid metabolism pathways similar to humans [18,19]. For example, hamster plasma contains cholesteryl ester transfer protein, absent in rats and mice, which can transfer HDL-C to LDL-C when cholesterol levels are elevated. In addition, hamsters have similar enzymatic pathways in the lipoprotein and bile metabolism. Therefore, this study evaluated the preventive effect of fermented lemon juice (FLJ) supplementation on blood lipid regulation by lipid regulation using hyperlipidemic hamsters.

The purpose of this study is to explore the impact of fermentation engineering and its cycle management on the benefits of FLJ. In addition, the study investigated if FLJ had a hypolipidemic effect and identified its effects on hamsters fed a high-fat diet (HFD) to establish if it may effectively prevent obesity.

## 2. Materials and Methods

### 2.1. Preparation of the Lemon Juice and Fermentation

Contracted farmers supplied organic green lemons from the southern part of Taiwan. After washing, the lemon juice was extracted by squeezing the whole lemons, including the peel and seed. To prepare the FLJ, the extracted lemon juice was inoculated with cultivated DMS32004 and DMS32005 isolated from organic lemons (Jian Mao Biotechnology Co., Ltd., Kaohsiung City, Taiwan). The yeast concentration was 5 × 10^6^~5 × 10^7^ CFU/mL and fermentation was conducted at 28 °C at a pH of 2.3. After 21 days of fermentation, the FLJ was sterilized at 90 °C for 15 min and stored in a sealed container at room temperature until used.

### 2.2. Determination of the Juices’ TPC

A previously reported method [20] was used to examine the unfermented lemon juice and FLJs’ TPC. In brief, 10 mL of the unfermented lemon juice and FLJ were centrifuged. The sediment was removed, and the remaining juice was used as the sample. Then, 1 mL of sample was added to a 100 mL quantitative flask containing at least 60 mL of distilled water, and 5 mL of Folin–Ciocalteu reagent was added; the mixture was allowed to stand for 8 min. Subsequently, 15 mL of 20% sodium carbonate solution and 100 mL of distilled water were added; the mixture was allowed to stand for 1 h. The absorbance of the mixtures was measured using a BioTek Eon microplate spectrophotometer (Agilent Technologies Inc., Santa Clara, CA, USA) at a wavelength of 760 nm.

### 2.3. Animal Care

The study was conducted according to the guidelines of the Declaration of Helsinki, and approved by the Institutional Animal Care and Use Committee (IACUC) (protocol code MG-109328 and date of approval 13 November 2020). Fifty male Syrian hamsters were purchased from the National Laboratory Animal Center (Taipei, Taiwan). Animals were housed in the MedGaea Life Sciences Institute Animal Room (Medgaea Life Science Ltd., Taipei, Taiwan) in hamster cages under a 12-h light/dark cycle (6 a.m. light on and at 6 p.m. light off) at an air-conditioned temperature of 22 ± 3 °C. After one week, healthy hamsters were selected for the experiments. Food (LabDiet^®^ 5001; Purina Mills, Inc., Richmond, IN, USA) and purified water were provided ad libitum. The experimental hamsters were randomly divided into five equal groups (ten hamsters per group).

### 2.4. High-Fat Diet Composition

Hamsters, except the control group, were fed a HFD to establish an animal model of hyperlipidemia. The standard chow (LabDiet^®^ 5001) contained 3.36 kcal/g with 28.5% protein, 13.5% fat, and 58.0% carbohydrates. The HFD was standard chow, 91.7% (wt/wt), supplemented with cholic acid, 0.1% (wt/wt), cholesterol, 0.2% (wt/wt), lard oil, 3% (wt/wt), and soybean oil, 5% (wt/wt). The control group was fed standard feed, whereas the hyperlipidemia model hamsters were fed a HFD daily for six weeks to induce hyperlipidemia.

### 2.5. Remedy Designs

Animals were randomly assigned to five groups (*n* = 10 each) of similar average body weight as follows: control (standard chow), HFD without FLJ (HFD), HFD with low-dose FLJ (3.1 mL/kg/day), HFD with medium-dose FLJ (6.2 mL/kg/day), and HFD with high-dose FLJ (9.3 mL/kg/day) groups. The hamsters were housed, one per cage, in a controlled environment (22 ± 3 °C, 12-h light/dark cycle) with free access to food and water during the acclimatization and study periods. The hamsters were fed FLJ (only applied on the FLJ feeding groups) via an oral gavage from the first day of the test for 6 weeks. 

The hamster dose of FLJ was based on the US Food and Drug Administration’s human-equivalent dose to estimate the maximum safe starting dose in initial clinical trials for therapeutics in healthy adult volunteers. The recommended use of FLJ (which includes polyphenols and superoxide dismutase-like activity) for humans is about 25 mL daily for a normal diet. Therefore, assuming a human weight of 60 kg, the human equivalent dose would be (25 mL/day/60 kg); this equates to a hamster dose of 3.1 mL/kg/day using the conversion coefficient 7.4 to account for differences in the body surface area between hamsters and humans. The FLJ administered dose in this test is once (low-dose), double (medium-dose), and triple (high-dose) of the recommended human dose.

### 2.6. Data Collection

Collection of blood: At the end of 6 weeks, all animals were fasted for 12 h. The blood samples were obtained by cardiac puncture under isoflurane inhalation anesthesia. The fresh blood was collected, settled for 2 h, and centrifuged at 1200× *g* for 15 min to obtain the serum samples.

Collection of feces: Body weight was measured every week, and feces were collected during the final 3 days of the experiment for analysis.

Collection of tissue samples: The hamsters were sacrificed simultaneously at the end of the experimental period under isoflurane inhalation anesthesia, and the heart, liver, spleen, and kidney were removed immediately. The organs were weighed after washing with normal cold saline and sucked up dry. The largest lobe of the liver was stored at −70 °C for further use.

### 2.7. Determination of Serum Total Cholesterol and Triglycerides

The hamster serum collected as described previously was assayed for levels of TC, TGs, using a BioTek Eon microplate spectrophotometer (Agilent Technologies Inc., Santa Clara, CA, USA).

Total triglycerides: Serum (10 μL) was combined with 1 mL of the Triglyceride FS 5′ multipurpose kit (cat no. 1 5760 99 10 023; DiaSys Diagnostic Systems GmbH, Holzeim, Germany) at 37 °C for 10 min. The resulting sample’s absorbance was determined at the optical density at 500 nm (OD500) and compared with the calibrator TruCal U (cat no. 5 9100 99 10 064; DiaSys Diagnostic Systems GmbH, Holzeim, Germany) using the following formula: Triglyceride (mg/dL) = (ODSample − ODblank)/(ODcalibrator − ODblank) × 135
where 135 mg/dL is the calibrator concentration and OD is the optical density at 500 nm.

Total cholesterol: Serum (10 μL) was combined with 1 mL of the Cholesterol FS 10′ multipurpose kit (cat no. 1 1300 99 10 024; DiaSys Diagnostic Systems GmbH, Holzeim, Germany) at 37 °C for 10 min. The sample’s absorbance was then determined at an OD500 and compared with the calibrator TruCal U (cat no. 5 9100 99 10 064; DiaSys Diagnostic Systems GmbH, Holzeim, Germany) using the following formula: Cholesterol (mg/dL) = (ODSample − ODblank)/(ODcalibrator − ODblank) × 155
where 155 mg/dL is the calibrator concentration and OD is the optical density at 500 nm. 

### 2.8. Determination of Serum Lipoprotein Cholesterol Concentrations

The centrifugation method specified by the Taiwan Food and Drug Administration (Evaluation method for regulating blood lipid function of healthy food, no. 0960403114) was used to determine the serum samples’ very-low-density lipoprotein cholesterol (VLDL-C), LDL-C, and HDL-C. First, 0.5 mL of serum was added to 2.5 mL sodium bromide (NaBr) (density (D) = 1.006 g/mL), and the samples were centrifuged at 453,000× *g* for 3.5 h at 4 °C; 0.5 mL of the supernatant’s top fraction, VLDL-C (D ≤ 1.006 g/mL), was collected. Then, 0.5 mL NaBr (D = 1.230 g/mL) was added, and the sample was centrifuged at 453,000× *g* for 3.5 h at 4 °C. Then, 1 mL of the supernatant’s top fraction, LDL-C (1.006 g/mL < D ≤ 1.063 g/mL), was collected. Finally, 1.5 mL NaBr (D = 1.406 g/mL) was added before centrifuging at 453,000× *g* for 3.5 h at 4 °C; the top fraction was HDL-C (1.063 g/mL < D ≤ 1.210 g/mL). The various lipoproteins were measured using a Cholesterol FS 10′ multipurpose kit (cat no. 1 1300 99 10 024; DiaSys Diagnostic Systems GmbH, Holzeim, Germany).

### 2.9. Determination of Total Cholesterol and Triglycerides in the Liver

The protocol of Folch et al. [21] was used to determine the TC and TGs in the hamsters’ livers. First, chloroform: methanol (2:1, *v*/*v*) (TEDIA Chloroform, cat no. CS1332-001; Tedia Company Inc., Fairfield, OH, USA; Macron Fine Chemicals^™^ Methanol, lot no. 000209176; Avantor Inc., Radnor, PA, USA) was used to homogenize the liver tissue. The homogenized sample was then centrifuged at 3000× *g* for 10 min, and the supernatant was collected. Following the protocol of Carlson and Goldfarb [22], 0.9% saline was added to the homogeneous lipid liquid, and it was mixed well before being centrifuged at 1200× *g* for 5 min. The upper supernatant layer was removed, keeping the lipid phase layer, and it was placed in an oven at 95 °C until the organic solvent was volatilized. The resulting material was dissolved and mixed with lipid liquid (tert-butyl alcohol:Triton X-100:methanol, 2:1:1) (Sigma-Aldrich^®^ tert-butyl alcohol, lot no. SHBJ9404; Sigma-Aldrich^®^ Triton^™^ X-100, lot no. SLBN2536V, both Merck KGaA, Darmstadt, Germany). The livers’ TC and TGs were measured using a Cholesterol FS 10′ multipurpose kit.

### 2.10. Determination of the Total Cholesterol and Triglycerides in the Feces

The protocol of Folch et al. [21] was used to extract the fecal lipids, which were extracted using chloroform: methanol (2:1, *v*/*v*) to homogenize the tissue. The homogenized samples were centrifuged at 3000× *g* for 10 min, and the supernatant was collected. Then, following the protocol of Carlson and Goldfarb [22], 0.9% saline was added to the homogeneous lipid liquid. The solution was mixed well and then centrifuged at 1200× *g* for 5 min. The upper layer of the supernatant was removed, and the lipid phase layer was placed in an oven at 95 °C until the organic solvent was completely volatilized. The remaining residue was dissolved and mixed with lipid liquid (tert-butyl alcohol: Triton X-100: methanol, 2:1:1). The TC and TGs in the feces were measured using a Cholesterol FS 10′ multipurpose kit.

### 2.11. Statistical Analysis

All data were expressed as mean ± standard deviation (SD). Significant differences were established with one-way analysis of variance and Duncan’s multiple range test. Statistical significance was considered at *p* < 0.05.

## 3. Results

### 3.1. Lemon Juice Fermentation Evaluation

First, the total phenol content of the control unfermented lemon juice and the FLJ were compared to identify the optimal fermentation time. The results showed that the total phenol content on the 21st and 28th days of fermentation were significantly higher than that of the unfermented lemon juice (Figure 1).

### 3.2. Changes in the Experimental Hamster’s Body and Organ Weights

During the test period, the test animals’ activity, coat color, and reactions were normal, and there were no cases of hair loss, abnormal clinical symptoms, or death. There was no significant difference in the body weight of the experimental animals in each group (Table 1 and Table 2). In terms of the relative weight of the liver (g/100 g body weight), the HFD group’s liver weight was significantly higher compared with the control group (*p* < 0.05) (Table 1 and Table 2).

Furthermore, the results showed that the relative weight of the liver in each FLJ dose group significantly differed from that of the HFD group (*p* < 0.05). These results suggested that FLJ could reduce fat accumulation in the liver. During the test period, except for the control group, there was no significant difference in the average daily food intake between the low-dose, medium-dose, and high-dose FLJ groups and the HFD group (Table 3 and Table 4).

### 3.3. Total Cholesterol, Triglycerides, and Lipoprotein Levels in the Serum

After the experimental period of 6 weeks, the serum TG and TC values of each FLJ dose group were significantly lower compared with the HFD group (*p* < 0.05), as shown in Table 5.

The HDL-C and LDL-C values in the serum are shown in Table 6. The HDL-C of each experimental group was not significantly increased compared with the HFD group (*p* > 0.05). However, the LDL-C of each FLJ dose group was significantly lower compared with that of the HFD group (*p* < 0.05). These results suggest that FLJ can effectively reduce the blood’s TC, TG, and LDL-C concentrations.

The elevated ratio of HDL-C/LDL-C is a negatively correlated risk factor for atherosclerosis disease [23,24], and the elevated ratio of TC/HDL-C is the most efficient cardiovascular risk predictor by several studies; in general, the higher the ratio, the higher the risk [25,26]. Compared with the HFD group, the HDL-C/LDL-C ratio of each FLJ dose group was significantly increased (*p* < 0.05). However, the TC/HDL-C ratios of all the FLJ dose groups were significantly lower compared with the HFD group (*p* < 0.05) (Table 7).

### 3.4. Total Cholesterol and Triglyceride Levels in the Liver

The test results showed that the TC and TG contents in the HFD group’s livers were significantly higher compared with those of the control group (*p* < 0.05). Compared with the HFD group, the livers’ TC and TG concentrations were significantly decreased compared with each FLJ dose group (*p* < 0.05) (Table 8).

### 3.5. Total Cholesterol and Triglyceride Levels in the Feces

At the end of the sixth week, the TC and TG levels in the feces of each FLJ dose group were significantly higher than those of the HFD group (*p* < 0.05) (Table 9).

## 4. Discussion

Chronic diseases are major health problems faced by most countries. It is generally believed that excessive total calories, alcohol, and fat intakes, lack of exercise, and disorders of the metabolism are the leading causes of hyperlipidemia [27]. Common lipid-lowering drugs can be divided into four types: hydroxymethylglutaryl-CoA reductase inhibitors (statins), fibric acid derivatives (fibrates), niacin (nicotinic acid), and bile acid sequestrants; however, these drugs have many side effects [28]. The common side effects of statins medication include headache, dizziness, nausea, muscle pain, constipation, and indigestion [29,30,31]. Fibrates may cause dyspepsia, dizziness, abdominal pain, constipation, and gallstones. Nicotinic acid side effects include rapid heartbeat, itching, flushing, nausea, and vomiting [32]. The side effects of bile acid sequestrants include myopathy, increased liver enzymes, nausea, heartburn, loss of appetite, and indigestion [33,34]. Previous studies have shown that fermented orange juice attenuates dietary glycemic and triglyceride responses [35]. Therefore, diet and preventing lipid deposition in blood vessels and the liver may provide a solution for improving this chronic syndrome [36,37,38].

The literature shows that bioflavonoids in citrus may prevent neovascular diseases. For example, bergamot (Citrus bergamia) juice significantly reduces serum cholesterol, triglycerides, and LDL, and increases serum HDL levels [39]. The flavonoids in lemons, including hesperidin, eriocitrin, naringin, and narirutin, are often present in glycosylated forms. Among them, naringenin has been shown to have an anti-inflammatory effect and hypolipidemic activity, while naringenin has a hypocholesterolemic effect [40,41,42]. Flavonoid substances are often degraded by heat treatment. In contrast, the fermentation process can enhance the production of bioactive extractable phytochemicals, ensuring that heat-treated products retain bioactive components and improve bioavailability [43].

This experiment evaluated the blood lipid regulation effect of FLJ in hamsters with HFD-induced hyperlipidemia. Unlike the typical fermentation with sugar and a long fermentation time, this study used organic lemon raw materials and selected specific strains DMS32004 and DMS32005 of bacteria to shorten the fermentation time without added sugar, improving the fermentation efficiency through fermentation management effects. The fermentation cycle evaluation demonstrated that the total phenolic content of the FLJ was 0.754 mg/mL on the 21st day of fermentation and 0.781 mg/mL on the 28th day, and the total phenolic content of the unfermented lemon juice was 0.620 mg/mL; furthermore, the total phenolic content was higher than the unfermented lemon juice. The 21-day fermentation was chosen for this study based on the fermentation evaluation results. Therefore, the fermentation process has been reported to produce higher levels of bioactive extractable phytochemicals [11,12].

Elevated cholesterol or triglyceride levels can easily cause vascular endothelial cell dysfunction. In addition, lipoproteins can freely enter and exit the blood vessel walls when their concentration in the blood is too high. Moreover, lipoproteins can accumulate in the inner layer of the arterial vessel walls, causing local inflammation and attracting monocytes to adhere to the blood vessels’ inner layer, differentiating them into macrophages, which devour oxidized LDL-C to form foam cells. The death and accumulation of foam cells formed by macrophages, coupled with the proliferation and repair of connective tissue, may create early fatty streaks. If arteriosclerosis continues, then atherosclerotic plaque will be formed. Macrophages also secrete some cytokines, stimulate the proliferation of smooth muscle cells on the vessel wall, cause plaque fibrosis, accelerate arteriosclerosis, make the vessel lumen smaller, and make blood flow difficult [44,45].

In addition to causing heart disease, hyperlipidemia is closely related to chronic conditions, such as strokes, high blood pressure, diabetes, and kidney disease. The LDL-C/HDL-C ratio is reported to be a more valuable biomarker than LDL-C or HDL-C levels alone, especially for predicting the risk of multiple diseases. If the ratio is low, atherosclerosis risk factors are reduced [46,47,48]. Based on the experimental results obtained from the sugar-free fermentation method and fermentation cycle, the hypolipidemic effect of FLJ on the hyperlipidemic hamster model was investigated. This study’s data after six weeks showed that FLJ could effectively reduce the concentration of TC, TGs, and LDL-C in blood. Compared with the negative control group, the ratio of HDL-C/LDL-C in each FLJ group was significantly increased, while the TC/HDL-C ratio was significantly reduced (*p* < 0.05).

A normal liver contains 4–5% of the liver weight as fat. If the fat content exceeds 10–15% of the liver weight, it is called fatty liver disease, typically an increase in oil in the whole liver [49]. More than half of patients with hyperlipidemia also suffer from fatty liver. In addition, patients with high triglycerides are more likely to suffer from fatty liver compared with those with high cholesterol. Fatty liver is generally divided into two categories according to different causes, namely alcoholic and nonalcoholic [50,51]. Hyperlipidemia and obesity are also common risk factors for nonalcoholic fatty liver disease [52]. Some animal studies have shown that a high-cholesterol diet with hesperidin can reduce triglycerides in blood lipids and increase cholesterol content in feces [53]. The HFD group was fed a high-fat and high-cholesterol feed, which resulted in excessive fatty liver accumulation. Therefore, the relative liver weight (g/100 g body weight) of the HFD group was significantly increased compared with the control group. In this study, the liver’s TC and triglyceride levels showed that FLJ effectively reduced the blood’s TC, TG, and LDL-C concentrations.

After six weeks of FLJ supplementation, the feces from all hamster groups were collected and analyzed for TC and TGs. The test results showed that the contents of TC and TGs in the feces of each FLJ dose group were significantly higher than those in the HFD group. Therefore, supplementing FLJ can promote the excretion of fecal TC and TGs.

This study demonstrated the sugar-free fermentation method and fermentation cycle management of lemon juice. The potential use of FLJ without added sugar in regulating blood lipids was investigated. Future research and verification will be carried out to isolate specific substances from the sugar-free FLJ and examine their mechanism of action.

## 5. Conclusions

FLJ was extracted from the whole lemons (including peel and seeds), which not only benefits from whole fruits with discarded peel and seeds nutrients, but also reduces agricultural waste and improves economic efficiency. In addition, unlike the typical sugar-added fermentation, FLJ is produced in a sugar-free fermentation, can prevent excessive hidden sugar consumption. This study showed that FLJ could reduce the content of TC, TGs, LDL-C, and TC/HDL-C in blood, reduce the content of TC and TGs in the liver, improve the HDL-C/LDL-C ratio, and promote the excretion of TC and TGs in the feces. Therefore, based on this study’s results, the effect of fermentation engineering and its cycle management on fermentation benefits can optimize fermentability. The results of the study confirmed that FLJ produced by the sugar-free fermentation method and the fermentation evaluation method had the effect of regulating blood lipids in hyperlipidemic hamsters.

## Figures and Tables

**Figure 1 nutrients-15-05089-f001:**
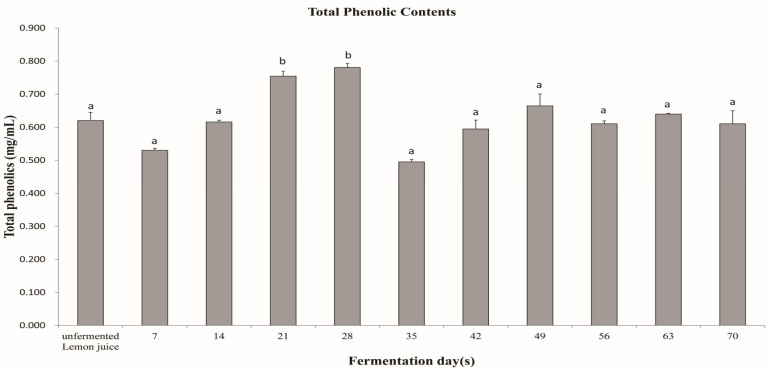
Total phenolic content of the unfermented and fermented lemon juice over time (days). Data are mean ± SD. Values with different superscript letters in the same row are significantly different, *p* < 0.05, by the one-way analysis of variance and Duncan’s multiple range test post hoc tests.

**Table 1 nutrients-15-05089-t001:** Weeks 0–4 average body weights of the treatment groups.

Groups	FLJ Dose (mL/kg)	Body Weight (g)				
		Week 0	Week 1	Week 2	Week 3	Week 4
Control	-	114.90 ± 15.69 ^a^	113.77 ± 15.29 ^a^	121.1 3± 13.38 ^a^	125.91 ± 12.48 ^a^	130.52 ± 12.81 ^a^
HFD	-	114.80 ± 14.90 ^a^	112.20 ± 16.21 ^a^	121.11 ± 14.14 ^a^	127.75 ± 14.43 ^a^	133.79 ± 14.24 ^a^
Low-dose	3.1	112.74 ± 3.30 ^a^	111.59 ± 12.31 ^a^	119.54 ± 12.55 ^a^	125.73 ± 10.91 ^a^	130.53 ± 12.47 ^a^
Medium-dose	6.2	114.59 ± 12.99 ^a^	111.96 ± 13.23 ^a^	121.60 ± 12.55 ^a^	128.25 ± 11.59 ^a^	133.64 ± 12.30 ^a^
High-dose	9.3	116.46 ± 11.77 ^a^	113.48 ± 13.23 ^a^	122.56 ± 10.18 ^a^	129.86 ± 11.05 ^a^	133.99 ± 11.42 ^a^

Week 0: Body weight on day 0 at the start of the trial. Data are mean ± SD, *n* = 10 hamsters per group. Values with different superscript letters in the same row differ significantly (*p* < 0.05). FLJ, fermented lemon juice; HFD, high-fat diet.

**Table 2 nutrients-15-05089-t002:** Weeks 5 and 6 average and the initial and final body weights of the treatment groups.

Groups	FLJ Dose (mL/kg)	Body Weight (g)				Liver Relative Weight Percentage (g/100 g BW)
		Week 5	Week 6	Before Fasting	Fasted	
Control	-	134.24 ± 13.48 ^a^	137.89 ± 13.41 ^a^	142.64 ± 14.16 ^a^	132.49 ± 13.27 ^a^	3.564 ± 0.297 ^a^
HFD	-	137.83 ± 13.57 ^a^	142.59 ± 14.15 ^a^	147.82 ± 14.01 ^a^	137.24 ± 13.83 ^a^	5.852 ± 0.498 ^a^
Low-dose	3.1	135.11 ± 12.85 ^a^	138.97 ± 11.12 ^a^	144.14 ± 10.98 ^a^	134.22 ± 10.77 ^a^	4.656 ± 0.206 ^a^
Medium-dose	6.2	138.39 ± 13.11 ^a^	142.67 ± 12.79 ^a^	147.38 ± 12.61 ^a^	137.78 ± 11.82 ^a^	4.556 ± 0.198 ^a^
High-dose	9.3	138.16 ± 12.01 ^a^	142.61 ± 12.53 ^a^	145.78 ± 13.23 ^a^	136.61 ± 12.31 ^a^	4.546 ± 0.228 ^a^

Data are mean ± SD, *n* = 10 hamsters per group. Values with different superscript letters in the same row differ significantly (*p* < 0.05). BW, body weight; FLJ, fermented lemon juice; HFD, high-fat diet.

**Table 3 nutrients-15-05089-t003:** Weeks 0–3 average daily food intake of the treatment group.

Groups	FLJ Dose (mL/kg)	Food Intake (g/day)		
		Week 1	Week 2	Week 3
Control	-	8.23 ± 0.91 ^a^	9.03 ± 0.67 ^b^	9.13 ± 0.90 ^b^
HFD	-	7.43 ± 1.24 ^a^	8.50 ± 0.94 ^a^	8.18 ± 0.59 ^a^
Low-dose	3.1	7.31 ± 1.06 ^a^	7.96 ± 0.72 ^a^	7.86 ± 0.69 ^a^
Medium-dose	6.2	7.53 ± 0.81 ^a^	7.90 ± 0.62 ^a^	7.80 ± 0.86 ^a^
High-dose	9.3	7.66 ± 0.93 ^a^	8.06 ± 0.64 ^a^	7.56 ± 0.65 ^a^

Data are mean ± SD, *n* = 10 hamsters per group. Values with different superscript letters in the same row differ significantly (*p* < 0.05). FLJ, fermented lemon juice; HFD, high-fat diet.

**Table 4 nutrients-15-05089-t004:** Average daily food intake of the treatment group: weeks 4–6.

Groups	FLJ Dose (mL/kg)	Food Intake (g/day)		
		Week 4	Week 5	Week 6
Control	-	9.22 ± 0.83 ^b^	9.00 ± 0.65 ^b^	9.16 ± 0.96 ^b^
HFD	-	8.39 ± 0.79 ^a^	8.24 ± 0.92 ^a^	7.81 ± 0.86 ^a^
Low-dose	3.1	7.81 ± 1.19 ^a^	8.03 ± 0.45 ^a^	7.81 ± 0.60 ^a^
Medium-dose	6.2	7.91 ± 0.65 ^a^	7.94 ± 0.64 ^a^	7.84 ± 0.38 ^a^
High-dose	9.3	7.70 ± 0.82 ^a^	7.80 ± 0.62 ^a^	7.60 ± 1.00 ^a^

Data are mean ± SD, *n* = 10 hamsters per group. Values with different superscript letters in the same row differ significantly (*p* < 0.05). FLJ, fermented lemon juice; HFD, high-fat diet.

**Table 5 nutrients-15-05089-t005:** Serum total cholesterol (TC) and triglyceride (TG) values in the treatment groups.

Groups	FLJ Dose (mL/kg)	TC (mg/dL)	TGs (mg/dL)
		Week 6	Week 6
Control	-	104.15 ± 12.34 ^a^	61.62 ± 10.42 ^a^
HFD	-	268.37 ± 22.17 ^c^	209.04 ± 24.44 ^d^
Low-dose	3.1	218.48 ± 28.4 ^b^	131.08 ± 52.38 ^c^
Medium-dose	6.2	213.7 ± 12.81 ^b^	104.76 ± 34.85 ^b^
High-dose	9.3	210.89 ± 22.96 ^b^	97.3 ± 37.2 ^b^

Data are mean ± SD, *n* = 10 hamsters per group. Values with different superscript letters (a–d) in the same row differ significantly (*p* < 0.05). FLJ, fermented lemon juice; HFD, high-fat diet.

**Table 6 nutrients-15-05089-t006:** Serum lipoprotein cholesterol values of the treatment groups.

Groups	FLJ Dose (mL/kg)	VLDL-C (mg/dL)	LDL-C (mg/dL)	HDL-C (mg/dL)
		Week 6		
Control	-	14.02 ± 3.73 ^a^	20.25 ± 3.39 ^a^	69.79 ± 11.05 ^a^
HFD	-	64.26 ± 6.80 ^c^	107.67 ± 10.37 ^c^	94.14 ± 11.54 ^b^
Low-dose	3.1	54.47 ± 11.51 ^b^	70.52 ± 13.85 ^b^	93.33 ± 8.24 ^b^
Medium-dose	6.2	53.56 ± 7.87 ^b^	66.43 ± 10.81 ^b^	93.47 ± 8.17 ^b^
High-dose	9.3	51.90 ± 11.55 ^b^	63.31 ± 12.95 ^b^	90.06 ± 12.31 ^b^

Data are mean ± SD, *n* = 10 hamsters per group. Values with different superscript letters (a–c) in the same row differ significantly (*p* < 0.05). FLJ, fermented lemon juice; HFD, high-fat diet; LDL-C, low-density lipoprotein cholesterol; HDL-C, high-density lipoprotein cholesterol; VLDL-C, very-low-density lipoprotein cholesterol.

**Table 7 nutrients-15-05089-t007:** Serum lipoprotein cholesterol ratios and total vs. high-density lipoprotein cholesterol ratios of the treatment groups.

Groups	FLJ Dose (mL/kg)	HDL-C/LDL-C	TC/HDL-C
		Week 6	Week 6
Control	-	3.555 ± 0.971 ^c^	1.505 ± 0.123 ^a^
HFD	-	0.880 ± 0.119 ^a^	2.868 ± 0.234 ^c^
Low-dose	3.1	1.366 ± 0.273 ^b^	2.339 ± 0.222 ^b^
Medium-dose	6.2	1.416 ± 0.322 ^b^	2.310 ± 0.260 ^b^
High-dose	9.3	1.510 ± 0.524 ^b^	2.402 ± 0.555 ^b^

Data are mean ± SD, *n* = 10 hamsters per group. Values with different superscript letters (a–c) in the same row differed significantly (*p* < 0.05). FLJ, fermented lemon juice; HFD, high-fat diet; LDL-C, low-density lipoprotein cholesterol; HDL-C, high-density lipoprotein cholesterol; TC, total cholesterol.

**Table 8 nutrients-15-05089-t008:** Total cholesterol (TC) and triglyceride (TG) levels in the livers of the treatment groups.

Groups	FLJ Dose (mL/kg)	Liver	
		TC (mg/g)	TG (mg/g)
Control	-	1.651 ± 0.506 ^a^	1.763 ± 0.975 ^a^
HFD	-	41.598 ± 4.148 ^d^	30.866 ± 7.467 ^c^
Low-dose	3.1	26.989 ± 4.664 ^c^	15.187 ± 4.324 ^b^
Medium-dose	6.2	24.821 ± 6.206 ^c^	17.890 ± 2.458 ^b^
High-dose	9.3	19.312 ± 2.491 ^b^	15.484 ± 3.706 ^b^

Data are mean ± SD, *n* = 10 hamsters per group. Values with different superscript letters (a–d) in the same row differed significantly (*p* < 0.05). FLJ, fermented lemon juice; HFD, high-fat diet.

**Table 9 nutrients-15-05089-t009:** Fecal triglycerides (TGs) and total cholesterol (TC) levels in the feces.

Groups	FLJ Dose (mL/kg)	Feces	
		TC (mg/g)	TGs (mg/g)
Control	-	1.038 ± 0.150 ^a^	1.272 ± 0.116 ^a^
HFD	-	6.281 ± 0.748 ^b^	1.611 ± 0.201 ^b^
Low-dose	3.1	8.359 ± 1.105 ^c^	2.136 ± 0.190 ^c^
Medium-dose	6.2	9.095 ± 0.998 ^d^	2.358 ± 0.297 ^cd^
High-dose	9.3	9.830 ± 1.463 ^c^	2.584 ± 0.299 ^d^

Data are mean ± SD, *n* = 10 hamsters per group. Values with different superscript letters (a–d) in the same row differ significantly (*p* < 0.05). FLJ, fermented lemon juice; HFD, high-fat diet.

## Data Availability

The data presented in this study are available on request from the corresponding author.

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
