# Peer review of "Optimized Sugar-Free Citrus Lemon Juice Fermentation Efficiency and the Lipid-Lowering Effects of the Fermented Juice"

_nutrients, 2023, doi:10.3390/nu15245089_

Round 1

Reviewer 1 Report

Comments and Suggestions for Authors

Manuscript ID: Nutrients-2736570

“Optimized Sugar-free Citrus lemon juice fermentation Efficiency and the lipid-lowering effects of the fermented juice”

This study demonstrated the sugar-free fermentation method of lemon juice. The use of FLJ without added sugar in regulating blood lipids in hamsters was investigated. The questions were carefully created and technically discussed.

A few questions are:

1.      The introduction would benefit from an expanded list of references to reinforce the study's objectives. For example, in lines 62-63, where the statement "Studies suggest that the intake of citrus fruits and their juices can prevent cardiovascular disease, which may be related to citrus bioflavonoids" is made, it is advisable to include the corresponding reference. Furthermore, on line 291, the study should specify the particular side effects associated with commonly used lipid-lowering drugs.

2.      While there are existing reports on polyphenols in both mice and humans, the current study should elucidate its unique contributions and novel discoveries. What sets this study apart from previously published research on the same subject?

3.      In line 235, the authors assert that FLJ has the potential to diminish fat accumulation in the liver. To enhance the clarity of this claim, additional details supporting how FLJ achieves this effect should be provided.

4.      Could you elaborate on the significance of the TC/HDL-C ratio? What insights or implications does it offer in the context of the study?

Reviewer 2 Report

Comments and Suggestions for Authors

This study ficused on the effect of fermented lemon juice on high fat diet in hamster.

Major points :

Abstract: To long! should be re-written in smouth manner withour repetition.

Introduction: Authors didn't really explain why they chose to use hamsters unstead of mouse for tis study.

M&M:

- Remedy Designs: how lemon juice has been administrated to hamsters... by gavage? if yes, what controls received?

- Describe the method for the Total phenolic coentent. The unfermented  lemon juice was made fresh or stored? in which condition?

Table 1. what's the meaning of "a" behind values? there is no statistical significance between all values.

Table 2. the same question as for table 1. And, Liver relative percentage was clacualted in fasted animals?

Table 3; the title is not correct. You mean food intake?  So the corrdct title should be : Weeks 0–4 average food intake of the treated groups.

Table 4: is food intake.

I would suggest to make a graph assembing then resukts for Tabales 3 and 4.

Table 5. please give the exact statistical significance for each letter: a, b c, d.

Regarding the effect on liver lipid metabolism, the authors should evaluate the expression of gene markers for triglyceride, cholesterol and lipoproteins metabolism in the liver.

Author Response

請參閱附件。

Round 2

Reviewer 1 Report

Comments and Suggestions for Authors

Concerns were addressed properly. 

Reviewer 2 Report

Comments and Suggestions for Authors

Auhors answered most of the reviewer' concerns.